# Seasonal Variation in Aflatoxin Levels in Edible Seeds, Estimation of Its Dietary Intake and Vitamin E Levels in Southern Areas of Punjab, Pakistan

**DOI:** 10.3390/ijerph17238964

**Published:** 2020-12-02

**Authors:** Ahmad Faizal Abdull Razis, Muhammed Muzammel Shehzad, Sunusi Usman, Nada Basheir Ali, Shahzad Zafar Iqbal, Nadia Naheed, Muhammad Rafique Asi

**Affiliations:** 1Natural Medicines and Products Research Laboratory, Institute of Bioscience, Universiti Putra Malaysia (UPM), Serdang 43400, Selangor, Malaysia; usunusi.bch@buk.edu.ng; 2Department of Food Science, Faculty of Food Science and Technology, Universiti Putra Malaysia (UPM), Serdang 43400, Selangor, Malaysia; nada44basher@gmail.com; 3Department of Applied Chemistry, Government College University Faisalabad, Faisalabad 38000, Pakistan; muzammelshehzad786@gmail.com (M.M.S.); eruditenadia@gmail.com (N.N.); 4Food Toxicology Lab, Nuclear Institute for Agriculture and Biology (NIAB), Faisalabad 38000, Pakistan; asimuhammad@yahoo.co.uk

**Keywords:** AFs, edible seeds, tocopherols levels, dietary intake

## Abstract

A total of 779 samples of edible nuts (melon seeds, watermelon seeds, pumpkin seeds, and cantaloupe seeds) from Southern Punjab (Pakistan), were collected during the summer and the winter seasons. The natural occurrence of aflatoxins (AFs) and vitamin E (tocopherols) levels were investigated using HPLC. The results have shown that 180 (43.4%) of samples from the winter season and 122 (33.4%) samples from the summer season were found positive for AFs. Elevated average levels of total AFs (20.9 ± 3.10 μg/kg, dry weight) were observed in watermelon seeds without shell, and the lowest average amount (15.9 ± 3.60 μg/kg) were documented in melon seeds without shell samples from the winter season. An elevated average amount of total AFs 17.3 ± 1.50 μg/kg was found in pumpkin seeds available without a shell. The results have documented a significant difference in total AFs levels in edible seeds available with shells versus without shells (α = 0.05 & 0.01). The highest dietary intake of 6.30 μg/kg/day was found in female individuals from consuming pumpkin seeds (without shell) in the winter season. A value of 3.00 μg/kg/day was found in pumpkin seed without shell in the summer season in female individuals. The highest total tocopherol levels were 22.2 ± 7.70 ng/100 g in pumpkin seeds samples from the winter season and 14.5 ± 5.50 mg/100 g in melon seed samples from the summer season. The variation of total tocopherol levels in edible seeds among the winter and summer seasons showed a significant difference (*p* ≤ 0.0054), except watermelon seeds samples with non-significant differences (*p* ≥ 0.183).

## 1. Introduction

In recent years, owing to their beneficial nutritional value linked with the management of acute diseases such as cardiovascular conditions, cancer, obesity, and diabetes, the use of seeds as a dietary additives has increased substantially [1,2]. The functional and nutraceutical properties are focused on their high content of essential proteins, fatty acids, synthetic fibers, antioxidants, carotenoids, minerals, and vitamins [2]. Seeds of the Cucurbitaceae family (including watermelon or pumpkin) are mostly thrown out, however, they can be utilized for food enrichment or nutraceutical development. [3]. Herbaceous plant chia seeds are considered as one of the most nutritious foods due to their bioactive peptides and proteins [4]. The majority of edible seeds are usually consumed as vegetable oils [5]. The seeds of pumpkin, winter melon, and watermelon are huge, plentiful, and edible, and pumpkin seeds are roasted with relish in some geographic areas [6]. Charmagaz (watermelon seeds, melon seeds, cantaloupe seeds, and pumpkin seeds) are extremely popular in Pakistan and believed to help in brain development and rejuvenation. They are mostly used to make different types of drinks such as sardie (a cold drink) and make sweet dishes, smoothies, consumed in summer and halwa.

The edible seeds may be affected due to bad weather conditions and poor pre-harvest or post-harvest conditions. The drying, storage, and transportation processes are critical and may contribute to fungal attacks [7]. Insect and pests are the main cause of damage, but molds are frequently identified as major contributors to agricultural goods’ damage during storage [8]. These fungi produce mycotoxins, which may have lethal and carcinogenic impacts on human and animal health [9].

Mycotoxins are recognized as naturally occurring secondary metabolites produced by filamentous fungi with various chemical structures [10,11]. The Food and Agricultural Organization (FAO) reported that globally some 25 percent of crops are infected with mycotoxins [12,13]. Major classes of mycotoxins with highest influence on human beings and agriculture losses are aflatoxins (AFs), ochratoxin A (OTA), zearalenone (ZEA), deoxynivalenol (DON), and fumonisin (FB) [14,15]. However, nuts and seeds are mostly contaminated with aflatoxins [16]. The most studied class of AFs is a group of compounds formed primarily by *Aspergillus flavus and Aspergillus parasiticus* [17,18]. Previous studies have shown that 20 different types of AFs are known, but the most commonly known classes are AFB_1_, AFB_2_, AFG_1_, and AFG_2_ [19,20]. Health effects like teratogenicity, mutagenicity and hepatocarcinogenicity are caused due to the fact these toxins most attack the liver [21]. The International Agency for Research on Cancer (IARC) has classified AFs as carcinogenic to humans (Group 1). The relative toxicity ranking of the various forms of AFs is AFB_1_ > AFG_1_ > AFB_2_ > AFG_2_ [22]. The most important and deeply studied class of AFs is AFB_1_ due to its high toxicity and widespread occurrence in various staple foods and feeds documented in previous studies. Almost 119 countries have established regulations for AFs in international trade and defined their regulatory limits [23]. The maximum acceptable limits for total aflatoxins (AFB_1_ + AFB_2_ + AFG_1_ + AFG_2_) are 15 μg/kg for hazelnuts, groundnuts, Brazilian nuts, apricot kernels, almonds, and pistachios for food practices [24]. However, no regulations are implemented for these toxins by Pakistan. 

Vitamin E, commonly occurring in various foods like edible oils, cereals, margarine, nuts, fatty fishes, and egg yolk, contains numerous essential lipophilic organic compounds necessary to maintain the human’s smooth functions body. It is also used as food additives because of their antioxidants properties and their ability to protect against fat rancidity [25]. Some studies have documented that, antioxidants were found effective in inhibiting or control aflatoxigenic fungi and the production of AFB1 and fumonisins in stored maize [26,27]. 

The environmental conditions of Pakistan are favorable for fungal propagation and, consequently, for the production of AFs [28]. The seasonal variation may affect aflatoxins contamination during storage or selling in the supermarket or open shops. The humid environment may increase moisture level and provide favorable conditions for fungal growth. In our previous studies, a considerable amount of AFs in dry fruits was documented (highest levels of total AFs 7.30 ± 1.80 μg/kg in peanuts without shell, the lowest mean of 2.90 ± 1.50 μg/kg in watermelon seeds with shell samples) [29,30]. The high amounts of AFs were observed in peanut and peanut products [31]. Therefore, the present research is aimed to examine; the natural presence of AFs levels in edible seeds, to compare the amounts of AFs in edible seeds with the European Union (EU) recommended limits, to determine the dietary intake of AFs in the local population, and to investigate the vitamin E content in selected edible seeds. The findings of the current work will be beneficial for farmers, consumers, and traders to create awareness about the health risks related to these toxins, e.g., to purchased edible seeds in proper packing and the seeds available in shells have fewer levels as compared to without shell samples. 

## 2. Materials and Methods

### 2.1. Sampling

A total of 779 samples of edible seed (melon seeds, watermelon seeds, pumpkin seeds, and cantaloupe seeds) samples (414 from the winter season and 365 from the summer season) were collected from various cities in Southern Punjab (Multan, Bhakkar, Layyah, and Muzaffargarh) Pakistan. The seeds samples were directly purchased from the market, open shops, and a superstore. The edible seed samples were obtained randomly. The terms of seasons, i.e., the summer season, is comprised of (May 2019 to August 2019) and the winter season (November 2019 to January 2020). The deshelling of edible samples was done from a market, where samples were purchased. The sample size of each seed was kept at 1 kg, and the samples were ground in fine particle size with a grinding mill (Retsch, Dusseldorf, Germany). The mill was cleaned properly after grinding each sample. After grinding, the samples were stored in plastic polyethylene bags and placed in the laboratory (Food Safety and Food Toxicology, Department of Applied Chemistry, GCUF) at −20 °C in a freezer. 

### 2.2. Chemicals and Reagents

Aflatoxin standards (in acetonitrile 1 μg/mL) and chemicals like methanol (HPLC grade), chloroform (HPLC grade), cupric carbonate (HPLC grade), anhydrous sodium sulfate (HPLC grade) were acquired from (Sigma-Aldrich, Steinheim, Germany). The other chemicals and reagents used in the current research were of high purity grade (≥90%), and the double-distilled water was used for analysis.

### 2.3. Aflatoxins Extraction 

The extraction of AFs from edible seeds was carried out following the method of Schuller and Van Egmond [32] with some modifications. The edible seeds were kept at 40 °C for dryness in a vacuum oven and then milled. Then, 40 g of dried ground edible seeds were taken and added in 200 mL of methanol/water (80:20 *v*/*v*) solution. Afterward, 5 g of NaCl was added to this mixture and blended for 2 min. Whatman no.1 filter paper was used to filter the mixture. After carrying out the filtration, 50 mL of the filtrate was taken in a separatory funnel, and 50 mL of chloroform was added, shaken gently for 1 min on the shaker, and left to develop phases. Then 5 g of cupric carbonate was taken in a beaker, and the aqueous chloroform layer was separated and filtered using anhydrous sodium sulfate. The chloroform extract was taken in a vial and dried using a nitrogen stream with controlled temperature (50 °C) after filtration. The samples were derivatized by adding 100 μL of trifluoroacetic acid (TFA) to 400 μL of mixture of samples and then vortexed for 30 s on vortex (Scilogex SCI-FS, Rocky Hill, CT, USA), and allowed the mixture to stand for 15 min. Lastly, the solution of 900 μL (water/acetonitrile) with a ratio of (9:1 *v*/*v*) was added and vortexed for 30 s and subjected for HPLC analysis. 

### 2.4. Extraction of Oil

The extraction of oil from each edible seed was determined following the method of Gliszczynska-Swigło et al. [33]. In a cotton thumb bell, 20 g of ground sample was taken, and soxhlet apparatus (at 110 °C, for a duration of 24 h) was used with 250 mL of n-hexane to extract the oil of edible seed sample. Approximately (0.04 to 0.12 g) of the sample was dispersed in 1 mL of 2-propanol and vortexed for 2 min and stored in a vial for HPLC analysis. 

### 2.5. HPLC Conditions

The analysis of AFs and tocopherols were performed on an HPLC instrument (Model-LC-10A, Shimadzu, Kyoto, Japan) equipped with a C_18_ column (250 mm × 4.6 mm, 5 μm) (Discovery HS, Bellefonte, PA, USA) provided with a fluorescence detector (RF-530). The mobile phase for tocopherols has consisted of 50% acetonitrile (solvent A) and 50% methanol (solvent B) with a 1 mL/min flow rate and a 20 μL of injection volume. The fluorescence detector was set at 325 nm of an emission wavelength and 295 nm of excitation wavelength. The flow rate of 1.5 mL/min of isocratic mobile phase with the composition of (water/methanol/acetonitrile) (60:20:20 *v*/*v*/*v*) was used for the analysis of AFs. The emissions and excitation wavelengths were 440 and 360, respectively. The final injection volume was 20 μL.

### 2.6. Dietary Intake Evaluation

According to the formula used by FAO/WHO [34], the estimated daily intake (EDI) can be calculated as:$$\mathrm{Estimated}\;\mathrm{Daily}\;\mathrm{Intake}\;\left(\mathrm{EDI}\right)\;\mu g/\mathrm{kg}/\mathrm{day}\;=\;\frac{\mathrm{Intake}\;\mathrm{rate}\;\mathrm{of}\;\mathrm{edible}\;\mathrm{seeds}\;\left(g\right)\times \;\mathrm{AFs}\;\mathrm{mean}\;\mathrm{level}\;\mu g/\mathrm{kg}\;}{\mathrm{Average}\;\mathrm{weight}\;\left(\mathrm{kg}\right)}$$

A frequency questionnaire was used to estimate the consumption data of edible seeds used in rice, sweet dishes, and cold drinks to approximately 650 participants and asking them about the edible seeds utilized in the last 4 weeks. The questionnaire was completed by considering all aspects of the intake of seeds, their dietary supplements, and consistency. The exact quantity of edible seeds used in food products was evaluated from the survey. The male and female participants had an average weight of 71.5 and 50 kg, respectively.

### 2.7. Statistical Analysis

The data were analyzed as triplicate replicates, and the mean levels were given as standard deviations. The seven-point standard curve was constructed, a straight-line equation was obtained, and the coefficient of determination (R^2^) was determined using linear regression. The significant differences in AFs levels in edible seeds from the summer and the winter seasons were determined using one-way ANOVA (α = 0.05) using SPSS (IBM Inc., Armonk, NY, USA).

## 3. Results and Discussion

### 3.1. HPLC Method Validation

The HPLC parameters were evaluated in terms of the limit of detection (LOD) and the limit of quantification (LOQ). The LOD and LOQ of AFB_1_ and AFB_2_ were 0.05 and 0.15 μg/kg, and 0.08 and 0.24 μg/kg, respectively, for AFG_2_ and AFB_2_. The LOD and LOQ were measured as the signal-to-noise ratio of S/N (3), as shown in Table 1. The precision and accuracy were calculated using recovery analysis. The recovery examination was carried out by adding 4, 8, and 20 μg/kg levels of AFB_1_ and AFG_1_, and levels of 2, 4, and 12 μg/kg of AFB_2_ and AFG_2_ were added to negative samples of mixed edible seeds. The current procedure has revealed a good recovery, ranging from 75 to 110% with RSD (relative standard deviation) from 10 to 19%. A previous study [29] reported recoveries of AFs in dry fruits and edible seeds samples varied from 83–90% with RSD 8 to 19%, comparable to the results of the present study. The values of LOD and LOQ for AFB_1_ and AFG_1_ were 0.04 and 0.12 μg/L and 0.06 and 0.18 μg/L for AFG_2_ and AFB_2_, respectively. The natural occurrence of individual AFB_1_, AFG_1_, AFG_2_, and AFB_2_ in the pumpkin seed sample is shown in Figure 1. 

### 3.2. Occurrence of AFs in Edible Seeds

The study was conducted for the examination of AFs in 414 samples of edible seeds (melon seeds, watermelon seeds, cantaloupe seeds, and pumpkin seeds) from the winter season and 365 samples of edible seeds from the summer season, and results are presented in Table 2. The maximum average of AFB_1_ and total AFs (16.5 ± 2.45 and 20.9 ± 3.10 μg/kg) was found without shelled watermelon seed samples from the winter season. Furthermore, the maximum average of AFB_1_ (14.4 ± 1.90 μg/kg) and total AFs (17.3 ± 1.50 μg/kg) was found without shell samples of pumpkin seed from the summer seasons. In shelled pumpkin seeds, the amount of AFs found in the summer season sample was 17.3 ± 1.50 μg/kg. In shelled watermelon seeds, the minimum level of AFB_1_ and total AFs were 6.70 ±1.90 and 8.4 ± 1.90 μg/kg, respectively, from the summer season. The findings showed that out of a total of 365 summer season experiments, 122 (33.4%) were aflatoxin-infected and 180 (43.5%) were aflatoxin poisoned out of 414 winter season experiments. The results have shown that 27.2% of samples of edible seeds having the levels of AFB_1_ higher than EU permissible limit (≥5 μg/kg), and 12.2% of samples having total AFs levels greater than ≥ 20 μg/kg from the summer season as presented in Figure 2. Furthermore, in 32.8% of samples levels of AFB_1_ greater than 5 μg/kg were found and 11.5% of samples from the winter season had a total amount of AFs higher than 20 μg/kg, as shown in Figure 3. 

In a previous study, Iqbal et al. [29] examined 320 samples of edible nuts (peanut, poppy seed, pistachio, almonds, cashew) and dry fruits (figs, plum, raisins, apricot, dates, watermelon seed, pomegranate seeds, and melon seeds) and observed that 128 (40%) samples were contaminated with AFB_1_ and the incidence of total AFs was comparable to the results of the current findings. The samples had levels of total AFs greater than 4 μg/kg and 10 μg/kg in 34 and 25% of samples, and the elevated average amount of total AFs (7.30 ± 1.80 μg/kg), was comparatively low to the results of the present study. In another study, Masood et al. [30] investigated 307 samples of dried fruits and edible nuts from Pakistan and observed that 132 (43%) of the samples were found to be contaminated with AFB_1_, and total AFs. The elevated average amount of total AFs i.e., 7.89 ± 0.99 μg/kg in peanuts without shell samples and the lowest average amount (2.45 ± 0.11 μg/kg) was observed in watermelon without shell samples. Huang et al. [35] have observed a very high percentage of samples from China to be contaminated with AFs, i.e., 93.9% of peanut butter, and documented an average amount ranging from 0.3 μg/kg to 95.9 μg/kg, a much higher range than the results of the present findings. Iqbal et al. [31] have analyzed 198 samples of peanut and peanut products from Pakistan and found that 61% of roasted peanut (shell samples), 68% of roasted peanut (without shell), 59%, of raw peanut (with shell), 55% of raw peanut (without shell), 50% of peanut butter, 42% of peanut cookies and 20% of peanut nemko were observed to be contaminated with AFs. The average concentration in raw peanut shell samples was 6.4 μg/kg, in roasted peanut with and without shell (10.4 and 12.3 μg/kg), in raw peanut without shell 9.6 μg/kg, peanut butter 2.4 μg/kg, peanut nimko (3.4 μg/kg), and in peanut cookies 4.6 μg/kg, much lower than the findings of the present study. A low percentage of AFs contamination and average amount have been documented by Kabak et al. for samples from Turkey [36]. The results demonstrated that among 300 samples of both dried figs and hazelnuts six (12%) of samples of hazelnut kernel (ranging from 0.09 to 11.3 μg/kg) and five (8.3%) samples of roasted hazelnut kernel (average 0.17 to 11.2 μg/kg) were found to be contaminated by AFs. Bankole et al. [37] analyzed 137 melon seed samples from Nigeria and documented a mean level of AFB_1_ (14.8 μg/kg) in forest melon seed and 11.3 μg/kg in savanna melon seeds. In another study, Juan et al. [38] analyzed 100 samples of dried fruits and nuts from Morocco and documented a very high mean level of AFB_1_ (2500 μg/kg) in walnut samples and an average level of 1430 μg/kg in pistachio samples, very high amounts compared to the results of the present findings. In the summer season, during the cultivation, the environmental conditions, especially the climatical variations, i.e., rainfall, broadly affects the growth of fungi in food products. Therefore, the weather and climatic conditions are considered important factors for AF production [39]. The fungal attack and levels of AFs in nuts and dried fruits may also vary during seasons. The months of July, August, and September are rainy periods with high rainfall levels and higher moisture and humidity levels [16,40]. The previous findings have shown that the most suitable temperature for the growth of various fungal species, particularly of *Aspergillus* species, ranges between 10.0 to 48.8 °C with 33.8 °C as an optimum temperature [38]. However, there was a significant difference in AFs levels in edible seeds from the shell and without shell samples (*p* < 0.05). 

### 3.3. Dietary Intake Estimation

The estimation of dietary intake in edible seeds from the summer and the winter season in male and female individuals is presented in Table 3. The results have shown that maximum dietary intake of AFs was assessed to be 4.38 μg/day/kg body mass (BW) in males for pumpkin seeds without shell from the winter season and the maximum amount of 6.3 μg/day/kg BW was found in winter season pumpkin seeds for female individuals. The minimum dietary intake was documented without shell melon seeds, i.e., 1.04 ug/day/kg BW in the winter period in males and from the summer season 0.59 μg/day/kg BW in males. In earlier findings, Heshmati et al. [41] confirmed that the dietary intake in dates, dried mulberries, figs, and apricots was 0.12, 0.04, 0.04, and 0.06 ng/kg/bw/day, respectively. Williams et al. [42] have documented lower levels of dietary exposure compared the current findings, i.e., levels of AFs was 11.4 to 158.6 ng/kg/day in Swaziland, 3.5 to 14.8 ng/kg/day in Kenya, 38.6 to 183.7 ng/kg/day in Mozambique, 16.5 ng/kg/day in Transkei (South Africa), 4 to 115 ng/kg/day in the Gambia, 11.7 to 2027 ng/kg/day from China, 6.5 to 53 ng/kg/day from Thailand, 2.7 ng/kg/day in the USA. Sugita-Konishi et al. [43], from Japan, have stated that the exposure for AFB_1_ ranged from 0.908 ng/kg BW/day to 0.909 ng/kg BW/day in one to six year-old children and 0.288 ng/kg BW/day to 0.289 ng/kg BW/day in adults of age more than 20 years, much lower than the current results. 

### 3.4. The Vitamin E Levels in Edible Seeds 

The variation of vitamin E levels in edible seeds from the winter and the summer season is presented in Table 4. 

The maximum average amount of total tocopherol content was found in melon seeds, i.e., 19.5 ± 4.90 mg/100 g, and the lowest total tocopherol content was observed in watermelon seeds 3.85 ± 3.60 mg/100 g from the winter season. However, the highest level of total tocopherol 14.5 ± 5.50 mg/100 g was found in melon seeds, and the most moderate mean level of 2.96 ± 5.60 mg/100 g was found in watermelon seeds from the summer season. The findings reveal the significant differences in vitamin E levels in samples from the winter and summer seasons (α = 0.05), except for watermelon seeds, which show non-significant differences (α = 0.05). The individual and total levels of vitamin E content in winter and summer seasons are presented in Figure 4.

The study has observed a negative correlation (Pearson correlation −0.370, and shown a significant difference between vitamin E and total AFs levels at α = 0.01) between AFs amount in different edible seeds and vitamin E content from winter and summer seasons, as shown in Figure 5. 

In a previous study Iqbal et al. [19] have shown significant differences (*p* < 0.05) in vitamin E contents in different rice varieties, and a negative correlation (r = −0.62) was found between AFs concentration and vitamin E levels, in agreement with the present findings. Previous studies have observed that selenium and retinol’s antioxidant properties, ascorbic acid, and tocopherol not only safeguard the membrane from the harmful effect of mycotoxins, but induce or boost the liver function to detoxify mycotoxin levels [44]. Furthermore, the extracts from certain medicinal herbs and plants could possibly defend from ochratoxin A, aflatoxin B1, and fumonisin B1 toxicity [45,46,47]. More comprehensive work to establish a relationship between AFs in edible seeds and variation in vitamin E content is recommended. 

## 4. Conclusions

The average amount of total AFs in edible seeds from the winter and summer seasons was considerably high compared to our previous studies. Furthermore, no significant difference in AFs levels was found in winter versus the summer season. However, there exists a significant difference in AFs variation in edible seeds samples consumed with shells versus without shell samples. The highest dietary intake of 6.30 μg/kg/day was calculated in female individuals from pumpkin seeds samples from the winter season. The highest vitamin E levels 22.2 ± 7.70 mg/100 g were found in pumpkin seeds from winter samples. There exists a negative correlation between AFs levels and vitamin E contents in edible seeds. The present research results should be informative for farmers, traders and consumers regarding the health consequences correlated with these toxins. We have purchased these edible seeds available in the market with shells or without shells, but in the future, we would be more interested in collecting samples during the field with fruits and then studying the pre-harvest content of fungi or aflatoxins.

## Figures and Tables

**Figure 1 ijerph-17-08964-f001:**
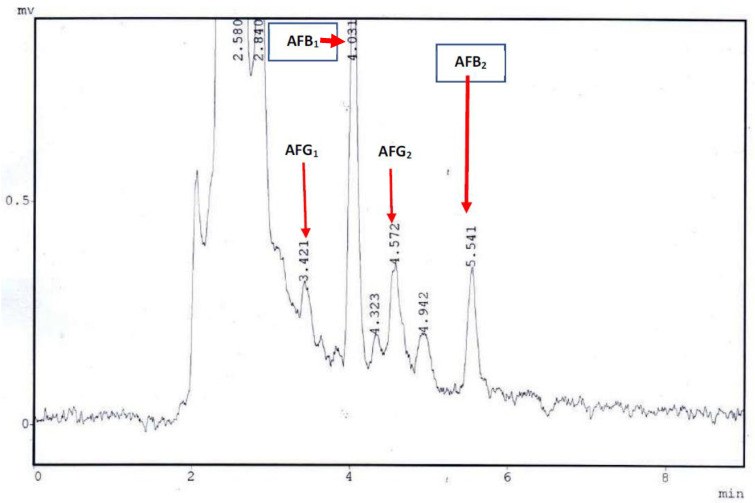
The natural occurrence of individual peaks corresponding to AFB_1_, AFG_1_, AFB_2_, and AFG_2_ in a pumpkin without shell seed sample.

**Figure 2 ijerph-17-08964-f002:**
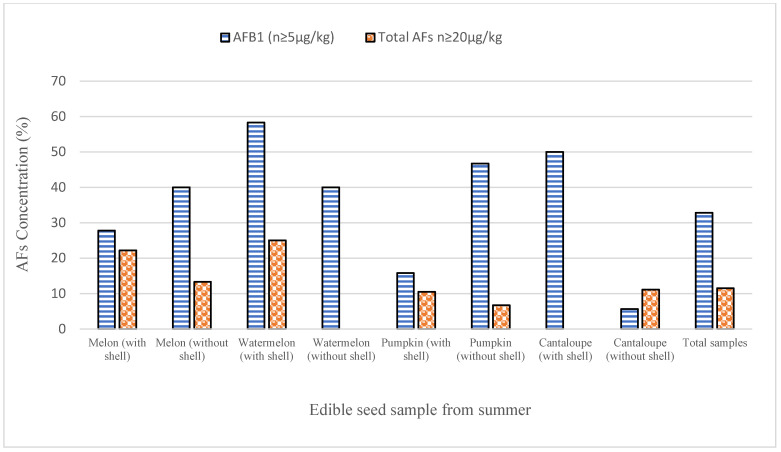
The number of AFB_1_ and total AFs (μg/kg) samples higher than the recommended EU limits from the summer season.

**Figure 3 ijerph-17-08964-f003:**
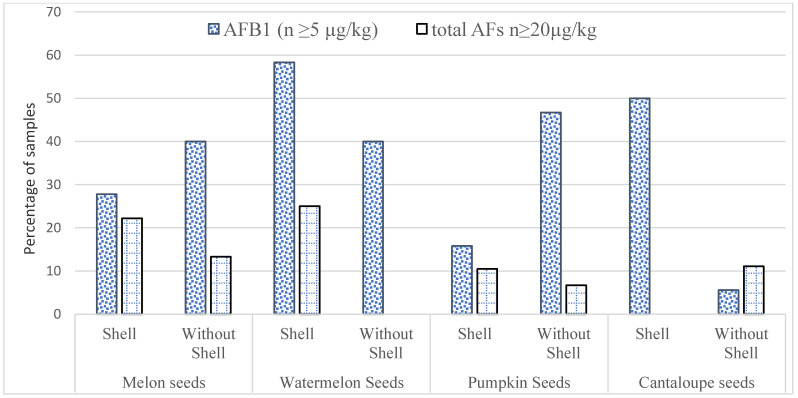
The number of AFB1 and total AFs (μg/kg) samples higher than the recommended EU limits from the winter season.

**Figure 4 ijerph-17-08964-f004:**
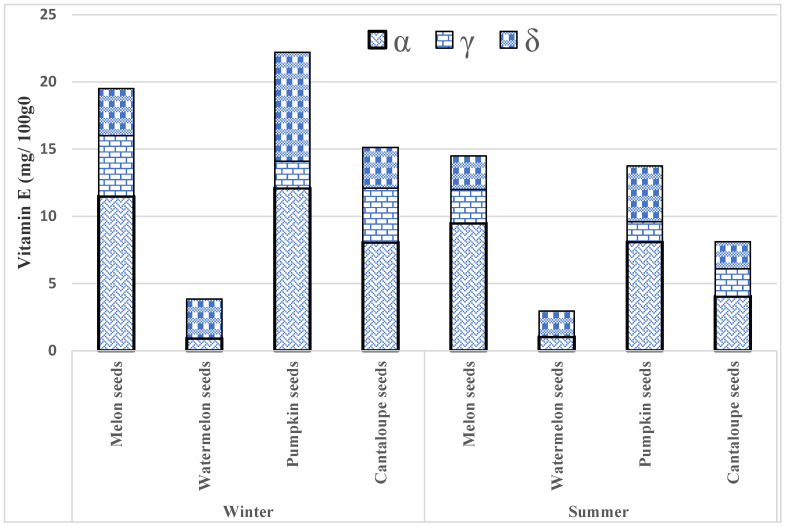
The variation of individual and total tocopherol in edible seed samples from winter and summer seasons.

**Figure 5 ijerph-17-08964-f005:**
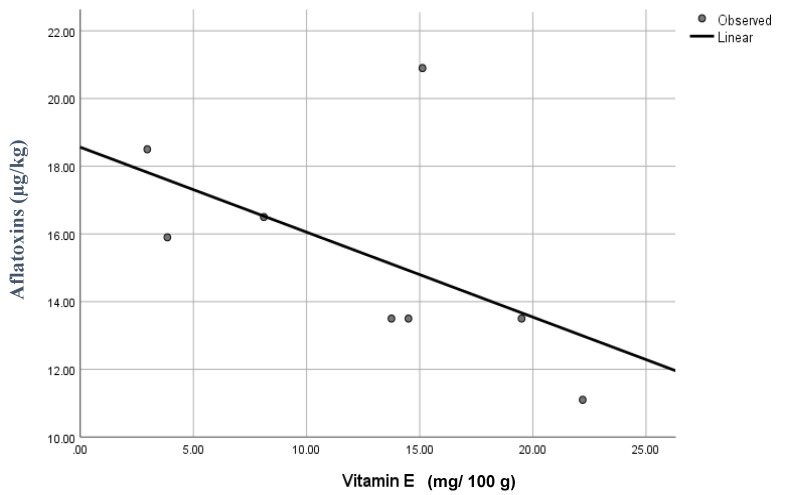
The correlation between vitamin E and total AFs concentrations.

**Table 1 ijerph-17-08964-t001:** Analytical parameters for the determination of AFs and tocopherols.

Aflatoxins	Linearity μg/mL	LOD μg/kg	LOQ μg/kg	R^2^	Precision (%RSD)	
					Reproducibility	Repeatability
AFB_1_	1–120	0.05	0.15	0.9981	17	15
AFB_2_	0.5–25	0.08	0.24	0.9947	14	16
AFG_1_	1–120	0.05	0.15	0.9972	12	17
AFG_2_	0.5–25	0.08	0.24	0.9972	10	16
α	0.5–60	0.04	0.12	0.9986	11	19
γ	0.1–30	0.07	0.21	0.9883	16	18
δ	0.1–30	0.07	0.21	0.9891	14	18

RSD: Relative Standard deviation; LOQ: Limit of Quantification; LOD: Limit of Detection. (α, γ, δ are the isomers of tocopherols).

**Table 2 ijerph-17-08964-t002:** Occurrence of AFB_1_ and total AFs (μg/kg) in edible seeds from summer and winter season from Punjab, Pakistan.

Seasons	Type	Melon Seeds		Watermelon Seeds		Pumpkin Seeds		Cantaloupe Seeds		Total
		Shelled	Without Shell	Shelled	Without Shell	Shelled	Without Shell	Shelled	Without Shell	
Winter Season	Total sample (*n*)	60	65	42	32	75	60	40	40	414
	Positive Sample *n* (%)	23 (38.33)	20 (30.76)	18 (42.85)	20 (62.5)	23 (30.66)	28 (46.66)	23 (57.5)	25 (62.5)	180 (43.4)
	AFB_1_ (μg/kg) ± SD	10.5 ± 2.10	12.6 ± 2.50	8.90 ±2.80	16.5 ± 2.45	12.9 ± 2.60	12.6 ± 3.40	7.9 ± 2.60	11.8 ± 3.20	
	Total AFs (μg/kg) ± SD	13.5 ± 3.40 *	15.9 ± 3.60 *	11.1 ± 2.10 **	20.9 ± 3.10 **	13.5 ± 2.90 **	18.5 ± 2.90 **	13.5 ± 3.10 *	16.5 ± 3.50 *	
	Range (μg/kg)	0.05–25.40	0.05–35.5	0.05–20.5	0.05–35.5	0.05–28.5	0.05–38.9	0.05–50.5		
Summer Season	Total sample (*n*)	50	60	40	40	60	50	30	35	365
	Positive Sample (%)	18 (36)	15 (25)	12 (30)	15 (37.5)	19 (31.7)	15 (30)	10 (33.3)	18 (51.4)	122 (33.4)
	AFB_1_ (μg/kg) ± SD	8.20 ± 2.50	10.9 ± 2.50	6.70 ±1.90	10.1 ± 1.90	9.8 ± 2.50	14.4 ± 1.90	6.3 ± 1.90	10.2 ± 2.80	
	Total AFs (μg/kg) ± SD	11.80 ± 2.10 *	13.2 ± 2.80 *	8.40 ±1.95 *	11.6 ± 1.80 *	15.0 ± 2.50 *	17.3 ± 1.50 *	10.5 ± 1.95 *	13.9 ± 2.10 *	
	Range (μg/kg)	0.05–25.5	0.05–39.8	0.05–23.6	0.05–45.6	0.05–29.8	0.05–39.7	0.05–20.5	0.05–33.2	

* = The significant difference of variation in total AFs levels in edible seeds available shelled versus without shell (α = 0.05). ** = The significant difference of variation in total AFs levels in edible seeds available shelled versus without shell (α = 0.01).

**Table 3 ijerph-17-08964-t003:** Estimation of dietary intake for AFs in edible seeds in the local population from Punjab, Pakistan.

Seasons	Type	Melon Seeds		Watermelon Seeds		Pumpkin Seeds		Cantaloupe Seeds	
		Shelled	Without Shell	Shelled	Without Shell	Shelled	Without Shell	Shelled	Without Shell
	Consumption g/day	20	5	10	20	15	10	15	10
Winter	AFs mean level (μg/kg)	13.5	14.9	11.1	13.5	20.9	18.5	13.5	14.5
	Dietary Intake (male) μg/kg/day	3.78	1.04	1.55	3.78	2.59	4.38	2.83	2.03
	Dietary Intake (female) μg/kg/day	5.4	1.50	2.20	5.40	3.70	6.30	4.10	2.90
Summer	AFs mean level μg/kg	11.8	12.20	8.40	11.60	17.30	15	10.50	11.90
	Dietary Intake (male) μg/kg/day	1.65	0.85	0.59	1.62	1.21	2.10	1.47	0.83
	Dietary Intake (female) μg/kg/day	2.40	1.20	0.80	2.30	1.70	3.00	2.10	1.20

Average Weight female = 55, Average age = 27.4. Average Weight male = 71.5, Average age = 29.8.

**Table 4 ijerph-17-08964-t004:** Detection of vitamin E content in edible seeds from Punjab, Pakistan.

Seasons	Seeds Type	Samples	α Tocopherol (mg/100 g)	γ Tocopherol (mg/100 g)	δ Tocopherol (mg/100 g)	Total Tocopherol Content (mg/100 g)
Winter Season	Melon seeds	125	11.5 ± 2.70	4.5 ± 2.30	3.5 ± 1.90	19.5 ± 4.90 *
	Watermelon seeds	74	0.93 ± 1.40	0.01 ± 4.15	2.91 ± 1.05	3.85 ± 3.60 ^N.S^
	Pumpkin seeds	135	12.1 ± 2.80	2.0 ± 2.50	8.1 ± 2.10	22.2 ± 7.70 *
	Cantaloupe seeds	80	8.07 ± 1.30	4.03 ± 1.15	3.02 ± 1.10	15.12 ± 4.65 *
Summer Season	Melon seeds	110	9.5 ± 3.75	2.5 ± 2.80	2.5 ± 2.94	14.5 ± 5.50 *
	Watermelon seeds	80	1.04 ± 2.45	0.01 ± 1.15	1.91 ± 1.45	2.96 ± 5.60 ^NS^
	Pumpkin seeds	110	8.1 ± 3.85	1.50 ± 2.10	4.15 ± 3.15	13.75 ± 6.50 *
	Cantaloupe seeds	65	4.05 ± 3.35	2.04 ± 3.10	2.02 ± 3.15	8.11 ± 3.40 *

* = The variation in total levels of tocopherols shows a significant difference in winter than summers seasons samples (α = 0.05). ^NS^ = The variation in total levels of tocopherols shows non-significant difference in winter compared to summers seasons samples (α = 0.05).

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
