# Peer review of "Seasonal Variation in Aflatoxin Levels in Edible Seeds, Estimation of Its Dietary Intake and Vitamin E Levels in Southern Areas of Punjab, Pakistan"

_ijerph, 2020, doi:10.3390/ijerph17238964_

Round 1
Reviewer 1 Report
ijerph-994707-peer-review-v1
Reviewer summary.
The goal of this work is to determine seasonal variation in aflatoxin levels of edible seeds (shelled v. deshelled), estimate dietary intake of edible seeds, and correlate vitamin E levels with occurrence and level of aflatoxin. The authors ‘randomly’ sampled 779 edible nuts (melon, pumpkin, watermelon, cantaloupe). The authors used HPLC to measure aflatoxin types and levels and different tocopherols present. I do understand that this study is a screening and survey-based study and there has been no data published on levels of edible seeds and exposure to aflatoxin in Punjab, Pakistan. However, I have major concerns with the lack of experimental description (see details below for specifics) and approach used to write this manuscript (details below, for example Figure 1 and Table 1 should be supporting data as they do not directly contribute to the objective of the study). Authors mention that the outcomes from this work will highlight the public health importance of aflatoxin contamination and create awareness but do not provide specifics on how and any background on the current regulations and safety standards in Pakistan or the region. I have highlighted some strengths in this study as well as some major comments and suggestions. I would also recommend major grammatical corrections throughout the text.
Strengths.
- Title of the manuscript accurately captures study goals and findings
- Strength in sample numbers (779), type of sample (edible seeds are not really studied for aflatoxin contamination), seasonal variation, deshelled v. shelled edible seeds, vitamin E levels
Major weaknesses, comments, and suggestions.
Introduction.
- Introduction and background is missing important components – I don’t see the link between vitamin E and aflatoxin
- I don’t see current regulatory and safety standards set for edible seeds in the region – how much of a public health concern this is?
- Lack of hypothesis on seasonal variation – did the authors predict that levels would be higher in one season v the other? What are possible explanations? Also, if samples are collected in a particular season, they may be stored and harvested in a previous season?
- Lack of benchmark for edible seeds – what is high and what is considered low? These are all relative and arbitrary – any data from previous studies on range of expected aflatoxin in seeds?
- I did not see any mention of how aflatoxin gets into these seeds? Soil? Do they concentrate in the seed vs. the flesh? So would people that consume melon and cantaloupe be also exposed to aflatoxin?
L72 – “to compare the amount of AFs in edible seeds with the European Union (EU) recommended limits” – authors should try to include these limits in the figures and tables so it makes it easy to compare to a benchmark level
L75 – “consumers and traders to create awareness, about the health risks related to these toxins”; what does create awareness mean? Provide examples and specifics – if the public is more aware of aflatoxin contamination, what can be done? Would authors propose local populations stop consuming these edible seeds?
Materials and Methods.
- No method or description on how seeds were deshelled (removal of shells)
L84 – …grinding mill (how did authors account for, did authors consider cross contamination of spores and aflatoxin here from one sample to next)?
L86 – 4C or freezer (-20C)?
L125 – frequency questionnaire (should be included in supporting); where is the metadata on survey results? Is it in an online repository? Lack of description on how data was collected from local population
Results.
- Table 1 and Figure 1 should go into supporting as the intent is to support the study - these data figure and tables do not directly provide data that support study objectives
- Figure 1 – part of the chromatogram is ‘cut off’, lack of description in figure legend
- Figure 2 – no error bars? where are standard error? statistical significance? figure legend missing important information and description
- L220-226; authors include these previous studies but do not provide context of how previous study relates to current findings
- L238-239; “The study has observed a negative correlation between AFs amount in different edible seeds and vitamin E content”; authors provide no interpretation of the data – how and why this was observed?
Discussion – the authors can significantly improve on discussion and interpretation of findings – I see minimal discussion presented in this manuscript.
Author Response
As attached in file

Reviewer 2 Report
Dear authors,
your research is very interesting but your presentation is not easy to read. Please give more information in methods and statistics. Please check the reference between the text and the ables. One figure is presented twicce. The text can be improved for unterstanding. The discussion of the results is still poor.
Best regards,
the reviewer

Author Response
The detailed responses are attached in word file

Reviewer 3 Report
I find the paper interesting, with a good experimental design and interesting results.
I have only several minor concerns:
- the correlation between the co-occurrence of AFs and tocoferols must be highlighted in the conclusion section.
- even if I am not an English native speaker, I suggest a deep revision of the manuscript.
Author Response
The responses are attached

Round 2
Reviewer 1 Report
Dear Authors,
Thank you for your quick response on this manuscript. I have reviewed your cover letter and additions to the manuscript, I believe that the paper has improved significantly - one minor comment, I would recommend authors include some future direction in discussion or conclusions including the response below.
1. Comment: I did not see any mention of how aflatoxin gets into these seeds? Soil? Do they concentrate in the seed vs. the flesh? So would people that consume melon and cantaloupe be also exposed to aflatoxin?
Response Its very good comments, unfortunately we have purchased these edible seeds available in market as shells or without shells, however, in future we would be more interested to collect samples during field with fruits and then study the effect of pre-harvest fungi or aflatoxins.
2. AFBI peak in chromatogram Fig 1. - could authors include arrow to specify peak - is it 4.031 min?
Stay safe and take care!
Author Response
The response has attached in file

Reviewer 2 Report
Dear authors,
the only comment: please show the data or a figure for correlation of AF and Vitamin E. After correction of a few minor errors ready for publication from my side.
Best regards,
the reviewer

Author Response
The details of comments are attached in file
